# Improvement of the Firocoxib Dissolution Performance Using Electrospun Fibers Obtained from Different Polymer/Surfactant Associations

**DOI:** 10.3390/ijms20123084

**Published:** 2019-06-24

**Authors:** Lauretta Maggi, Valeria Friuli, Enrica Chiesa, Silvia Pisani, Mirena Sakaj, Paolo Celestini, Giovanna Bruni

**Affiliations:** 1Department of Drug Sciences, University of Pavia, Via Taramelli 12, 27100 Pavia, Italy; lauretta.maggi@unipv.it (L.M.); enrica.chiesa01@universitadipavia.it (E.C.); silvia.pisani01@universitadipavia.it (S.P.); 2Department of Chemistry, Physical-Chemistry Section, University of Pavia, Via Taramelli 16, 27100 Pavia, Italy; mirena.sakaj01@universitadipavia.it (M.S.); giovanna.bruni@unipv.it (G.B.); 3Cosma S.P.A., Via Colleoni 15/17, 24040 Ciserano, Italy; p.celestini@cosma.it

**Keywords:** polymeric fibers, surfactant, electrospinning, dissolution rate, veterinary drug, poorly soluble drug

## Abstract

An electrospinning process was optimized to produce fibers of micrometric size with different combinations of polymeric and surfactant materials to promote the dissolution rate of an insoluble drug: firocoxib. Scanning Electron Microscopy (SEM) showed that only some combinations of the proposed carrier systems allowed the production of suitable fibers and further fine optimization of the technique is also needed to load the drug. Differential scanning calorimetry (DSC) and X-ray powder diffraction (XRPD) suggest that the drug is in an amorphous state in the final product. Drug amorphization, the fine dispersion of the active in the carriers, and the large surface area exposed to water interaction obtained through the electrospinning process can explain the remarkable improvement in the dissolution performance of firocoxib from the final product developed.

## 1. Introduction

Firocoxib is a non-steroidal anti-inflammatory drug for veterinary use that belongs to the class of selective COX-2 enzyme inhibitors. It is generally prescribed for the treatment of pain and inflammation, but is also used for postoperative pain and inflammation relief in orthopedic surgery. Firocoxib is used in veterinary medicine for its efficacy in long-term pain management and gastrointestinal safety, and can be administered daily for long periods [1,2,3,4].

The oral bioavailability of firocoxib may be limited due to its poor water solubility, in fact, oral absorption in dogs is 38% [5]. In order to increase the dissolution rate and/or solubility of this drug, an innovative technique has been used: electrospinning. Electrospinning is a technique used to prepare ultrafine fibers in a dimensional range from micro to nanometers with controlled surface morphology, shape, and porosity. The ultrafine fibers are generated through the application of a high and controlled electric field on a polymeric solution ejected through a needle [6].

The polymer solution is inserted into a syringe pump and a high voltage is applied through an electrode connected to the capillary. A thin jet of polymeric fluid (known as the Taylor cone) is formed when the electric field strength exceeds the surface tension and reaches the critical value of the electric potential [7], then, the solvent evaporates, producing the fibers. The main process parameters that can influence the morphology of the fibers are the intensity of the applied electric field, the flow rate, the solution physicochemical properties, the distance between the syringe and the collector, the collector type (plate or rotating drum), the needle dimension (gauge), and the environmental parameters (temperature, humidity, and airflow) [8].

Thanks to the unique properties of electrospun fibers such as high surface area, high porous mesh, high loading capacity, and simultaneous administration of diverse actives, this technique has mainly been applied in the production of carrier systems able to increase the dissolution rate of poorly soluble drugs for oral administration [9]. The polymer generally used for this purpose is polyvinylpyrrolidone (PVP), a water-soluble hydrophilic polymer. The electrospinning process promotes the molecular dispersion of the drug in the polymeric material and can also produce drug amorphization [10,11]. These two aspects are important in enhancing the drug dissolution process, but are not always sufficient to guarantee the release of the entire dose.

In this work, PVP fibers loaded with firocoxib were prepared, characterized, and tested in vitro to evaluate the improvement of the dissolution rate. The use of polyvinylpyrrolidone (PVP) showed an enhancement in the dissolution behavior of the drug, but did not allow the release of the entire dose. Therefore, other polymeric and non-polymeric materials were selected for association with the polyvinylpyrrolidone to improve the performance of the system. In particular, different surfactants were considered, those with a simple structure and those with a high molecular weight: sodium lauryl sulfate (SLS); two polyoxyethylene-polyoxypropylene block copolymers, poloxamer 188 (Lutrol^®^ F68) and poloxamer 407 (Lutrol^®^ F127); a polyvinyl caprolactam-polyvinyl acetate-polyethylene glycol graft copolymer (Soluplus^®^); and a hypromellose acetate succinate (Affinisol^®^ A716). This last material is a water-soluble polymer that can help keep the active ingredient stable in solid dispersions; moreover, it may inhibit the crystallization of the host molecules by promoting supersaturation of the drug in solution [12,13]. However, a high molecular weight polymeric carrier is needed to produce a network of coherent fibers through this technique. 

Thus, the aim of this work was to evaluate the possible associations of polymeric and non-polymeric materials to produce electrospun fibers able to improve the dissolution rate of firocoxib. SEM analysis was used to study the morphology of the fibers obtained with the different associations. The fibers were also characterized by differential scanning calorimetry (DSC), X-ray powder diffraction (XRPD), and Fourier infrared spectroscopy (FT-IR) techniques to define the solid phase of firocoxib in the fibers and the possible interaction among the components. Furthermore, the in vitro dissolution profiles of the drug from the carrier systems proposed were studied and compared to the physical mixtures of the same composition. The most promising system was optimized and its dissolution behavior was compared to the commercial product, Previcox (57 mg of firocoxib), which was used as a reference.

## 2. Results and Discussion

### 2.1. Electrospinning Process

Polymeric solutions were electrospun using optimized parameters to obtain a regular and homogeneous jet. All setup parameters are reported in Table 1.

The instrument parameter range was selected with a PVP polymeric solution (PVPel). Starting from 1 kV, the voltage value was increased bit by bit until there was a uniform jet formation. The dope flow-rate was fixed at 1 mL/h during this first optimization. The voltage value of 20 kV showed the best behavior in stretching the polymeric solution; the flow-rate was adjusted at 0.8 mL/h to reduce the solution amount at the needle’s tip, but no fibers could be obtained. 

The same parameters were used for the drug/polymer solution, F:PVPel (1:4), and the constant trend was jet maintained. Therefore, drug loading into the polymer solution did not result in affecting the electrospinning process. In the following, an extensive process optimization was conducted to obtain good fibers.

First, PVP solutions were prepared with SLS, PVP:SLSel (4:1), and two different Lutrol types, PVP:LF127el (1:1) and PVP:LF68el (1:1). The PVP:SLSel (4:1) sample during the electrospinning process made a discontinuous jet and dropped. PVP:LF127el (1:1) and PVP:LF68el (1:1) required a slight voltage increase (21 kV) to obtain a uniform jet, but beading events were observable on the fiber surfaces. For this reason, other parameter values were tested for samples PVP:SLSel (4:1), PVP:LF127el (1:1), and PVP:LF68el (1:1) (data not shown), but the solutions did not produce a homogeneous jet stretching. For this reason, in the following SEM analysis, only the samples obtained using the process parameters reported in the Table 1 were scanned.

Next, Affinisol A716 and PVP solutions without and with the drug PVP:A716el (1:1) and F:PVP:A716el (1:2:2), respectively, were electrospun using a reduced flow-rate of 0.3 mL/h. Regular and smooth fibers were collected on the aluminum foil using solution PVP:A716el (1:1) while the drug loaded solution, F:PVP:A716el (1:2:2), sometimes showed drip phenomena that ruined the matrix morphologies.

The electrospinning process of PVP in combination with Soluplus (SOL) and PVP:SOLel (1:1) required an increase in voltage up to 26 kV to obtain the formation of a stretch jet. The recovery on the plate collector displayed regular and defect free fiber deposition with both solutions used (PVP:SOLel (1:1) placebo and F:PVP:SOLel (1:2:2) drug loaded).

Two different polymers-drug ratios were tested. The F:PVP:SOLel (1:2:2) sample contained double the firocoxib concentration compared to the F:PVP:SOLel (1:4:4) sample. Both compositions resulted in homogeneous jet formation and good fiber production.

The SEM scans reported below confirmed what was seen during the electrospinning process.

### 2.2. Scanning Electron Microscopy

The samples PVPel, PVP:SLSel (4:1), and PVP:LF127 (1:1) showed a similar morphology (Figure 1). They were made of rounded particles with a diameter in the micrometer scale and few very thin fibers that resembled a cobweb. The sample PVP:LF68el (1:1) showed fine fibers randomly distributed with several beads. The sample PVP:A716el (1:1) was made of fibers with a very irregular diameter and were ribbon-shaped. The best fibers were those obtained with the sample PVP:SOLel.

For this last sample, the diameter distribution was determined by considering about 300 fibers and the normal-log function (Equation (1)) was chosen to fit the experimental data.
y = *a*·exp[−0.5(ln*x*/*x*_0_)/*b*^2^](1)

Figure 3 shows the histogram of the diameter distribution, the curve of the normal-log distribution, the mean value *x*_0_, the standard deviation *b*, and the *a* coefficient. The percentile values of the fibers diameters are listed in Table 2.

The photographs of the samples loaded with firocoxib are shown in Figure 2. Samples F:PVPel (1:4) and F:PVP:A716el (1:2:2) were characterized by the presence of beads, while the fibers made by the combination of the polymers PVP and SOL, with different percentages of firocoxib, were very homogeneous and regular, and no beads were present.

In Figure 3, the diameter distribution, the curve of the normal-log function, and the distribution parameters obtained for the loaded fibers made with the association of PVP and SOL are shown.

The parameters of the size distribution are reported in Table 2. The mean size of the fibers was lower when the drug was loaded. The very thin diameter, in the micrometric scale, is a great advantage to improve the surface area that is able to interact with water, and thus enhance the dissolution rate of the drug-carrier system.

### 2.3. Thermal Analysis

Firocoxib melts with a sharp DSC peak at 117.5 °C with an enthalpy change of 100.5 Jg^−1^. The plain polymeric fibers did not show any thermal event in the same temperature range of the drug. The DSC curves of the drug:polymer/s physical mixtures showed the melting peak of firocoxib with an enthalpy change in agreement with the value expected based on the system composition. In contrast, the same peak was absent in the DSC curve of all of the loaded fibers, suggesting that firocoxib is amorphous.

As an example, in Figure 4, the DSC curves of the pure drug and of the samples PVP:SOL (1:1), F:PVP:SOL (1:2:2), and F:PVP:SOL (1:4:4) are reported.

### 2.4. X-ray Powder Diffraction

The characteristic diffraction effects of firocoxib are present in its XRPD pattern (Figure 5, curve a) as expected for a crystalline compound, but are absent in the patterns of all of the loaded fibers where only the broadband of the polymers (Figure 5, curve b) is visible. As an example, the patterns of PVP:SOLel, F:PVP:SOLel (1:2:2), and F:PVP:SOLel (1:4:4) (Figure 5, curves c and d) are shown in Figure 5. This experimental evidence supports the DSC results, indicating that the drug is amorphous in the loaded fibers.

### 2.5. Infrared Spectroscopy

The FT-IR spectra of the pure firocoxib, PVP:SOL (1:1), F:PVP:SOL (1:2:2), and F:PVP:SOL (1:4:4) are shown in Figure 6. The spectra of the loaded fibers were very similar to the spectrum of PVP:SOlel and only the strongest absorptions of the drug, not overlapping those of the polymers, were visible, for example, the peak at 1765 cm^−1^ (C=O stretching) as a shoulder to the peak of the polymers at 1733 cm^−1^, the peak at 1149 cm^−1^ (SO_2_ symmetric stretching), and the peaks at 1114 cm^−1^ (C–O stretching) and 774 cm^−1^(aromatic ring). This is justifiable considering the system’s composition.

### 2.6. Solubility and In Vitro Dissolution Test

The firocoxib solubility measured after 2 h was very low: 4.77 ± 0.55 mg/L and the dissolution rate was also slow: the equilibrium solubility reached at 24 h was 19.58 ± 0.35 mg/L in deionized water after 25 °C. Both of these characteristics may adversely affect the drug availability for absorption. The dosage in dogs is 5 mg/kg [14].

Some associations were selected to study the effect of drug loading and the in vitro dissolution profiles were evaluated. The F:PVPel (1:4) fibers showed an increase in the dissolution rate of the drug compared to firocoxib alone, but the entire dose was not dissolved within 6 h. However, a simple mixture of the two components, F:PVPpm (1:4) did not show any improvement of the dissolution behavior of the drug (Figure 7).

The combination of PVP and hypromellose acetate succinate, F:PVP:A716el (1:2:2), did not show the expected results; in fact, the release rate and the percentage of drug dissolved were lower compared to those obtained by the fibers made only of PVP and the drug, F:PVPel (1:4). Moreover, the SEM image of F:PVP:A716el (1:2:2) showed fibers with uneven diameters and the presence of bulges (Figure 2), which could suggest a non-homogenous dispersion of the drug inside the fibers. Additionally, the physical mixture, F:PVP:A716pm (1:2:2), did not show a valuable improvement in the dissolution performances of the drug (Figure 7).

Only the fibers obtained by combining PVP with the polyvinyl caprolactam-polyvinyl acetate-polyethylene glycol graft copolymer, F:PVP:SOLel (1:2:2), showed a valuable improvement to the performance of the drug release profile, compared to the drug alone and to the commercial product, Previcox (Figure 8). 

The association of PVP with the polymeric surfactant, Soluplus, appeared as the optimal combination to enhance the dissolution performance of the firocoxib, most likely due to the synergic effect of the two products: PVP allows the molecular dispersion of the drug and its amorphization, and SOL promotes its wettability. 

The combination of PVP and SOL in the fiber allowed an increase of the drug dissolution rate, but the whole dose did not dissolve completely. For this reason, it was thought to further improve the effect of the two polymeric materials by increasing their weight in the system: F:PVP:SOLel (1:4:4). The dissolution profile of this new product showed the complete release of the entire dose of the loaded drug in a few minutes, while the simple mixture of the same components, in the same quantity, did not produce the same effect (Figure 8).

## 3. Materials and Methods

### 3.1. Materials

Firocoxib (F) was kindly donated by Cosma (Ciserano, Bergamo, Italy). Previcox^®^ (Merial, Lyon, France) tablets containing 57 mg of firocoxib was bought in a pharmacy and used as a commercial reference. 

Sodium lauryl sulfate (SLS) was supplied by Carlo Erba (Milano, Italy); poloxamer 188 and poloxamer 407, Lutrol^®^ F68 (LF68) and Lutrol^®^ F127 (LF127), respectively, and polyvinyl caprolactam-polyvinyl acetate-polyethylene glycol graft copolymer, Soluplus^®^ (SOL) was obtained through BASF (Ludwigshafen, Germany); polyvinylpyrrolidone, Plasdone^®^ K29/32 (PVP) was supplied by the GAF Chemical Corp. Wayne (Parsippany, NJ, USA); and hypromellose acetate succinate, Affinisol^®^ HPMC-AS 716 G (A716) was obtained through the Dow Chemical Company (Bomlitz, Germany).

The solvents used to solubilize the polymeric materials, the surfactants, and the active ingredients to prepare the solutions for the electrospinning process were dichloromethane (DCM), ethanol 96° (EtOH), and dimethyl sulfoxide (DMSO), which were all supplied by Carlo Erba (Milano, Italy). 

### 3.2. Methods

#### 3.2.1. Samples Preparation

Polymer solutions to be electrospun were prepared by dissolving the polymer PVP alone or in combination with the selected surfactants in a solvent blend of ethanol and dichloromethane in a 1:1 volume ratio (EtOH:DCM 1:1 *v/v*) and stirred for about 2 h at room temperature. DCM was selected for its solubilizing capacity and low boiling point (39.6 °C), while EtOH was chosen for its high dielectric constant value (24.3), which is able to make the solution conductive and responsive to the electrospinning electric field.

Firocoxib was previously dissolved in DMSO then added to the polymeric solutions in EtOH–DMC and stirred for about 2 h. The final drug concentration was 5% *w*/*v* for each preparation.

All sample preparations were sonicated (Sonica Ultrasonic Cleaner–Soltec) for 15 min before electrospinning to eliminate air bubbles.

The composition of all the solutions prepared are reported in Table 3.

The fibers were coded by adding the suffix -el. The physical mixtures (coded by adding the suffix -pm) were prepared by using a Turbula apparatus (W.A. Bachofen AG, Basel, Switzerland) with the same composition of the corresponding fibers. The physical mixtures were used as references to compare the results of the thermal and dissolution analyses.

#### 3.2.2. Electrospinning Process

Electrospinning set up NANON-01A equipped with a dehumidifier (MEEC instruments, MP, Pioltello, Italy) was used to obtain the electrospun fibers. Instrument parameters (voltage, flow-rate, needle-collector distance, needle size) were set up to obtain a homogeneous and continuous jet. A plane plate metallic collector covered with aluminum foil was used for the fiber collection. Temperature and humidity were kept under control constantly to improve solvent evaporation.

The electrospun nanofiber samples were stored in a drying chamber for two days to remove the residual solvents and then used for further analysis.

#### 3.2.3. Scanning Electron Microscopy

A Zeiss EVO MA10 (Carl Zeiss, Oberkochen, Germany) was used to analyze the morphology of the fibers. The samples were gold-sputter coated under argon to render them electrically conductive prior to microscopy.

#### 3.2.4. Thermal Analysis

Thermal characterization was carried out using a TGA Q2000 IR apparatus and a DSC Q2000 apparatus both interfaced with a TA 5000 data station (TA Instruments, NewCastle, DE, USA). The DSC instrument was calibrated using ultrapure (99.999%) indium (melting point = 156.6 °C; ΔH = 28.54 J g^−1^) as the standard. The calorimetric measurements were carried out in open standard aluminum pans under nitrogen flow (45 mL min^−1^) at 5 K min^−1^. All data from the thermal measurements are the average of three or more experiments.

#### 3.2.5. X-ray Powder Diffraction

XRPD measurements were performed using a D5005 Bruker diffractometer (Karlsruhe, Germany) (CuKα radiation, λ(Kα1) = 1.54056 Å; voltage of 40 kV, and current of 40 mA) equipped with a θ–θ vertical goniometer, Ni filter, monochromator, and scintillator counter. The patterns were recorded at room temperature in step scan mode (step size: 0.020, counting time: 3 s step^−1^) in the 3 < 2θ < 35 angular range.

#### 3.2.6. Infrared Spectroscopy

FT-IR spectra were obtained using a Nicolet FT-IR iS10 Spectrometer (Nicolet, Madison, WI, USA) equipped with an ATR (Attenuated Total Reflectance) sampling accessory (Smart iTR with diamond plate) by co-adding 32 scans in the 4000–650 cm^−1^ range at 4 cm^−1^ resolution.

#### 3.2.7. Solubility and In Vitro Dissolution Test 

Firocoxib solubility was determined using the shake-flask method in deionized water at 25 °C. An aliquot of the saturated solution was filtered (0.45 µm, Millipore). The drug concentration was determined by spectrophotometric detection at 275 nm (Lambda 25 UV-Vis, Perkin-Elmer, Monza, Italy) after 2 h (considered as the average transit time to reach the small bowel, which is the site of absorption [15,16]) and after 24–48 h to evaluate the equilibrium solubility [17].

Samples of firocoxib, loaded fibers (-el), physical mixtures (-pm), and the commercial product were evaluated in in vitro dissolution tests using the USP Apparatus 2 (Erweka DT-D6, Dusseldorf, Germany) paddle with a rotation speed of 100 rpm in 1000 mL of deionized water at 37 °C. The concentrations of the dissolved drug were determined on a filtered portion of the dissolution medium by UV detection at 275 nm (Lambda 25 UV, Perkin-Elmer, Monza, Italy). The data were processed through suitable software (Winlab V6 software, Perkin-Elmer, Monza, Italy) to obtain the dissolution profiles. All samples contained a dose of 57 mg of firocoxib and for each sample, six replicates were performed.

## 4. Conclusions

The electrospinning procedure can be optimized to produce fibers of micrometric size, with a different combination of polymeric and surfactant materials to load the drug in different ratios.

SEM analysis is an indispensable tool during production to refine the production process and to study the fibers’ dimensional output. DSC and XRPD indicates that the drug is in an amorphous state in the final product. A proper combination of different polymeric and non-polymeric material, the fine dispersion of the active ingredient in the carriers, and the large surface area exposed to water by the fibers of micrometric size can explain the remarkable improvement in the dissolution performance of firocoxib in comparison with the simple mixture of the same components. In particular, the association of the high molecular weight surfactant (polyvinyl caprolactam-polyvinyl acetate-polyethylene glycol graft copolymer) with a hydrophilic polymer, in this case study, allowed obtaining, at the same time, homogeneous fibers with a micrometric diameter, and an enhancement in the system wettability. 

In conclusion, the combination of PVP and SOL proved to be effective in promoting the dissolution rate of firocoxib from electrospun fibers, for this reason, the drug to carrier ratio was also optimized. The final product F:PVP:SOLel (1:4:4) showed the complete and fast release of the entire dose when compared to the commercial reference.

## Figures and Tables

**Figure 1 ijms-20-03084-f001:**
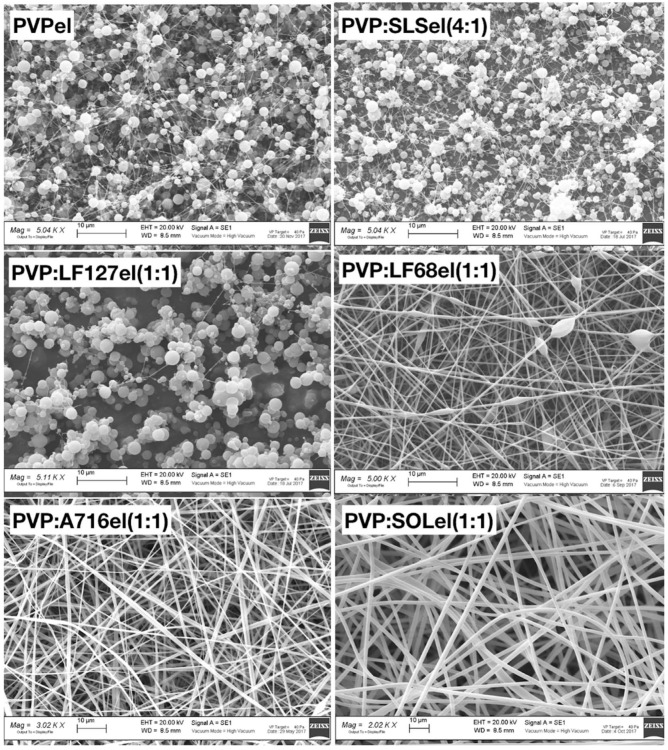
SEM micrographs of the unloaded polymeric fibers.

**Figure 2 ijms-20-03084-f002:**
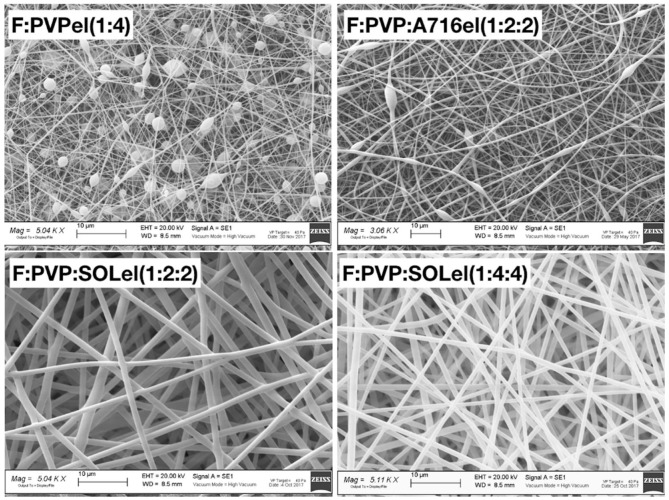
SEM micrographs of the polymeric fibers loaded with firocoxib.

**Figure 3 ijms-20-03084-f003:**
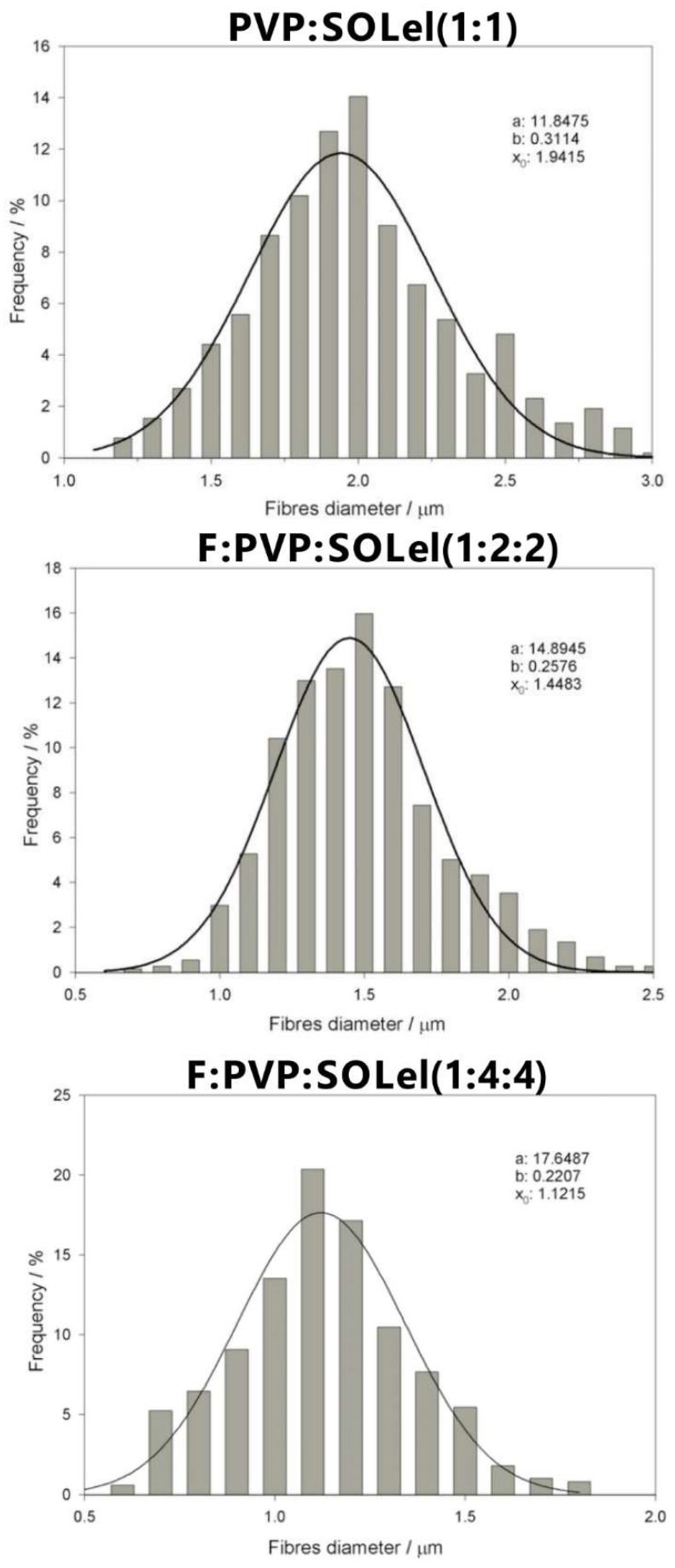
Histogram of the fiber size distribution and curve of the normal-log distribution, mean value, SD, and regression coefficient for PVP:SOLel (1:1), F:PVP:SOLel (1:2:2), and F:PVP:SOLel (1:4:4).

**Figure 4 ijms-20-03084-f004:**
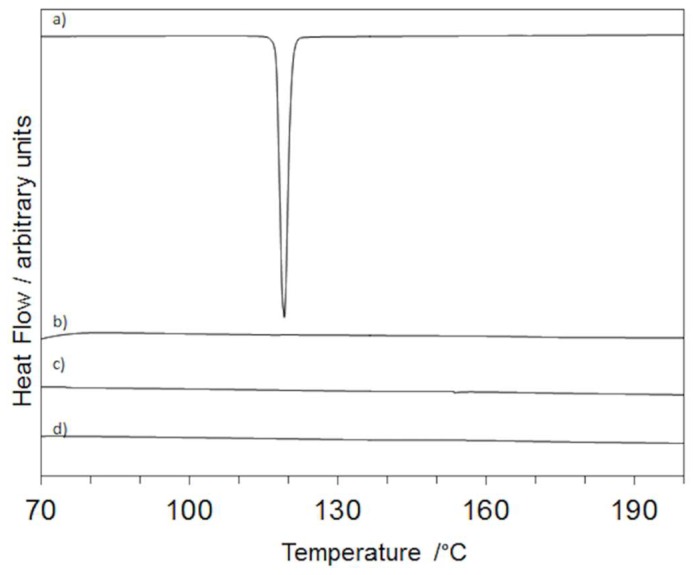
DSC curves of firocoxib (**a**), PVP:SOL (1:1) (**b**), F:PVP:SOL (1:2:2) (**c**), and F:PVP:SOL (1:4:4) (**d**).

**Figure 5 ijms-20-03084-f005:**
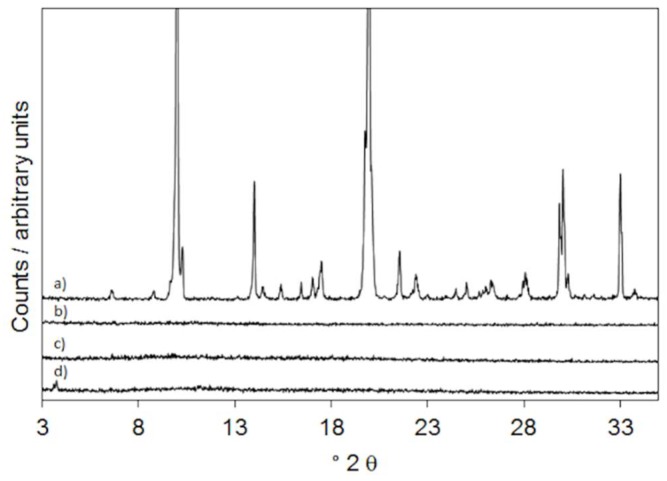
XRPD patterns of firocoxib (**a**), PVP:SOL (1:1) (**b**), F:PVP:SOL (1:2:2) (**c**), and F:PVP:SOL (1:4:4) (**d**).

**Figure 6 ijms-20-03084-f006:**
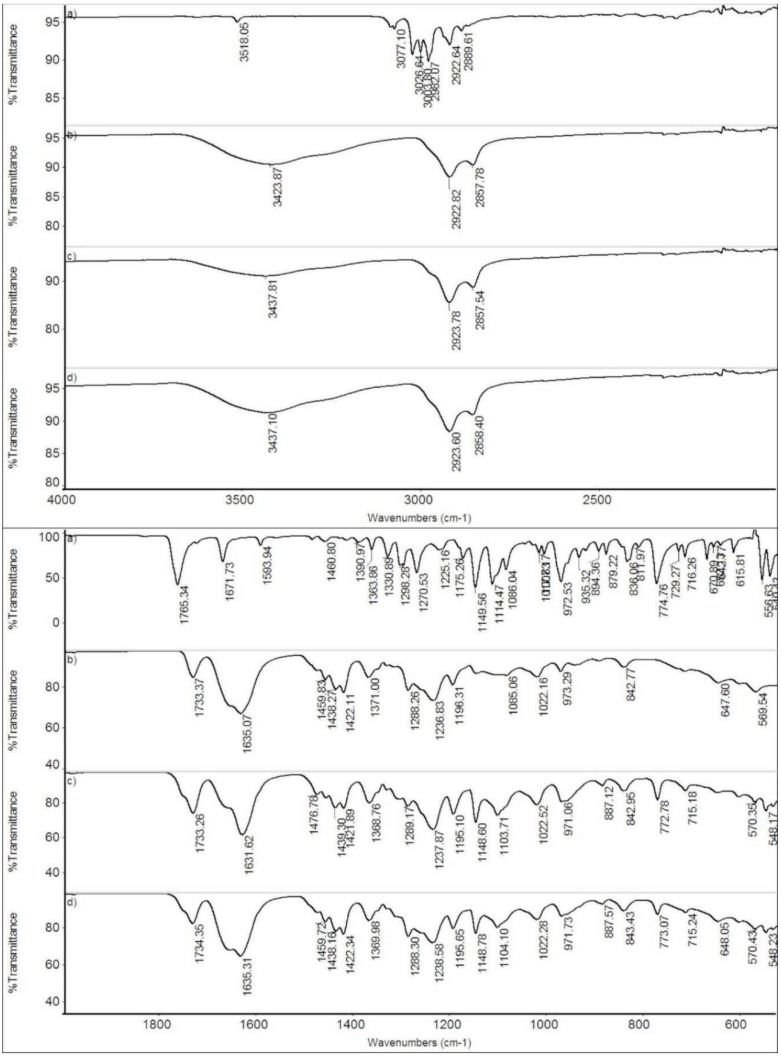
FT-IR spectra of firocoxib (**a**), PVP:SOL (1:1) (**b**), F:PVP:SOL (1:2:2) (**c**), and F:PVP:SOL (1:4:4) (**d**).

**Figure 7 ijms-20-03084-f007:**
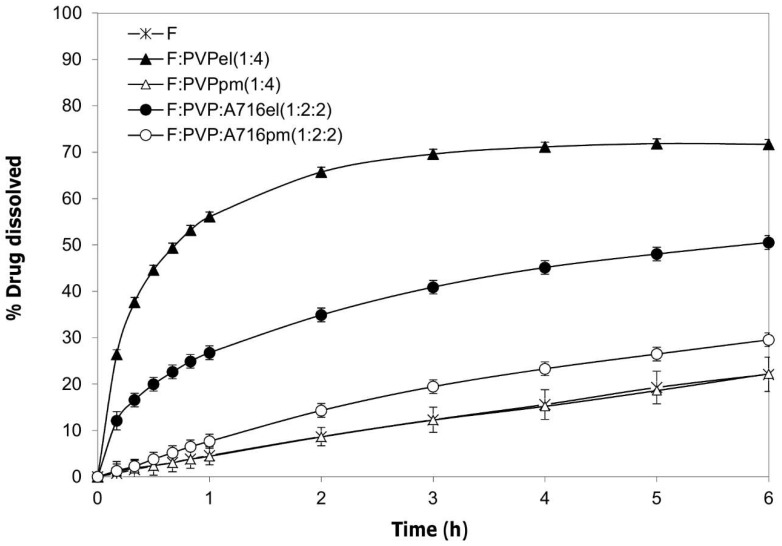
The dissolution profiles of the fibers made of PVP and F:PVPel (1:4) were compared to those obtained by combining PVP and hypromellose acetate succinate fibers, F:PVP:A716el (1:2:2), the corresponding physical mixtures, and firocoxib alone (F). All samples contained 57 mg of firocoxib.

**Figure 8 ijms-20-03084-f008:**
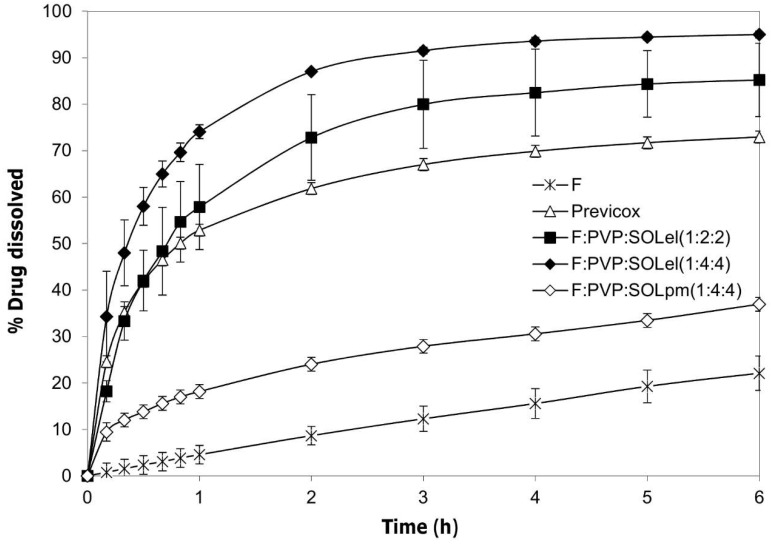
The carrier system containing a combination of PVP and the polymeric surfactant (polyvinyl caprolactam-polyvinyl acetate-polyethylene glycol graft copolymer) at two different concentrations. F:PVP:SOLel (1:2:2) and F:PVP:SOLel (1:4:4) were compared to the physical mixture F:PVP:SOLpm (1:4:4), to the commercial product, Previcox, and to firocoxib alone. All samples contained 57 mg of the active ingredient.

**Table 1 ijms-20-03084-t001:** Electrospinning process parameters optimized for each sample solution. The samples of the solutions prepared with firocoxib are reported in the last light grey rows.

Sample	Voltage (kV)	Dope Flow-Rate (mL/h)	Needle Collector Distance (cm)	Needle Diameter (G)	Temperature (°C)	Relative Humidity %	Fibers Forming
PVPel	20	0.5	15	18	30 ± 2	25 ± 2	beads
PVP:SLSel (4:1)	20	0.5	15	18	30 ± 2	25 ± 2	beads
PVP:LF127el (1:1)	21	0.5	15	18	30 ± 2	25 ± 2	beads
PVP:LF68el (1:1)	21	0.5	15	18	30 ± 2	25 ± 2	fibers/beads
PVP:A716el (1:1)	21	0.3	15	18	30 ± 2	25 ± 2	fibers
PVP:SOLel (1:1)	26	0.4	15	18	30 ± 2	25 ± 2	fibers
F:PVPel (1:4)	20	0.5	15	18	30 ± 2	25 ± 2	fibers/beads
F:PVP:A716el (1:2:2)	21	0.3	15	18	30 ± 2	25 ± 2	fibers/beads
F:PVP:SOLel (1:2:2)	26	0.4	15	18	30 ± 2	25 ± 2	fibers
F:PVP:SOLel (1:4:4)	26	0.4	15	18	30 ± 2	25 ± 2	fibers

**Table 2 ijms-20-03084-t002:** Percentile values of the electrospun fiber diameters: d(0.1), d(0.5), and d(0.9) are the fractions of fibers with the reported size or lower.

Sample	d(0.1)	d(0.5)	d(0.9)
PVP:SOLel (1:1)	1.54 µm	1.95 µm	2.34 µm
F:PVP:SOLel (1:2:2)	1.12 µm	1.45 µm	1.78 µm
F:PVP:SOLel (1:4:4)	0.84 µm	1.12 µm	1.40 µm

**Table 3 ijms-20-03084-t003:** Polymeric solutions prepared with and without firocoxib at different polymer concentrations (*w*/*v* %). The samples of the solutions prepared with firocoxib are reported in the last light grey rows.

Sample	Polymers *w*/*v* % Concentration in EtOH:DCM, 1:1						Firocoxib in DMSO
	PVP	SLS	Lutrol F127	Lutrol F68	Affinisol 716	Soluplus	
PVPel	20	-	-	-	-	-	-
PVP:SLSel (4:1)	20	5	-	-	-	-	-
PVP:LF127el (1:1)	10	-	10	-	-	-	-
PVP:LF68el (1:1)	10	-	-	10	-	-	-
PVP:A716el (1:1)	10	-	-	-	10	-	-
PVP:SOLel (1:1)	10	-	-	-	-	10	-
F:PVPel (1:4)	20	-	-	-	-	-	5
F:PVP:A716el (1:2:2)	10	-	-	-	10	-	5
F:PVP:SOLel (1:2:2)	10	-	-	-	-	10	5
F:PVP:SOLel (1:4:4)	20	-	-	-	-	20	5

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
