# Peer review of "Improvement of the Firocoxib Dissolution Performance Using Electrospun Fibers Obtained from Different Polymer/Surfactant Associations"

_ijms, 2019, doi:10.3390/ijms20123084_

Round 1

Reviewer 1 Report

 In the context of this manuscript an efficient solid formulation of the poorly water soluble COX-2 inhibitor NSAID, firocoxib, was developed. The technique they followed was based on the fabrication of firocoxib nanofibers using the electrospinning methodology. The aim of the authors to enhance that way the drug's aqueous release was successful. Manuscript, ID ijms-520391, is concisely written and well documented and it thus merits publication.  

Author Response

First of all, many thanks to the reviewers for their attention and their comments and suggestions.

Answers to the reviewer 1

English was professionally revised and some sentences were completely rewritten.

Reviewer 2 Report

The manuscript by Maggi et al. described the electrospinning of PVP/surfactant fibers and their characterizations of the fibers on an insoluble drug firocoxib. The manuscript is clearly written and could be an interest to readers who are investigating surfactants. However, there are several issues with the manuscript, and it is recommended the authors to correct the following items:

(1)In section 2.1 and 2.2, 20wt% of PVP solution alone typically is able to electrospin into fibers. Why beadings are there for the control PVP solution?

(2)In section 2.1 and 2.2, typically when poloxamers were added to polymer solutions, they tended to benefit electrospinning due to a change in the surface tension of the polymer solution. The 1/1 ratio of polymer/poloxamer is very significant amount, which can cause adverse effect in fiber morphology. However, the authors reported very uniform and consistent fiber formation. This leads to the question in (1) that the PVP control seemed to be incorrect.

(3)For table 1, it would be better if the authors add an additional column to show which formulation forms fibers.

(4)For table 2, it would be great if the control PVP solution forms fibers so the authors can compare the effect of poloxamer in fiber diameter. Generally speaking, the fiber diameter with poloxamer should be smaller since poloxamers benefit electrospinnability (better whipping) while the polymer concentration decreases.

(5)Figure 6 is recommended to improve its quality of presentation.

Author Response

First of all, many thanks to the reviewers for their attention and their comments and suggestions.

Answers to the reviewer 2:

1) You are right. PVP generally forms electro-fibers easily. But unfortunately in our conditions it did not. We suspect that it could be due to the combination of the solvents used. This particular combination was selected because it is able to dissolve all the components and in particular firocoxib. However PVP alone did not form good fibers using the electrospinning conditions that would permit firocoxib loading. For this reason, PVP was combined with other polymers that were selected on purpose to further enhance the drug solubility. Even a combination with a non-polymeric surfactant has been attempted, but it was not successful. However, in all the cases, an extensive process optimization was made to obtain regular fibers. Only two of these combinations lead to the formation of good fibers. 2.1 section was partially amended accordingly.

2) The presence of poloxamer slightly improved the formation of blank fibers. But it is the combination with the firocoxib solution in DMSO that make the difference and a remarkable improvement in the electrospinning output is evident.

What we want to underline that if we stopped the experiment at the beginning, when we tested only PVP, we could not reach our goal. Sometimes it is precisely the search for different combinations of polymers and solvents that could lead to the appropriate solution that was not so obvious from the beginning. Also the drug to carrier ratio could be particularly critical in the electrospinning process.

3) Table 1 was completed with the suggested new column.

4) You are right again, and the blank PVP fibers could be used as a comparison to study the process of fiber formation, but unfortunately we could not find a suitable combination of parameters to produce regular fibers.

5) Fig 6 quality has been improved.

English was professionally revised and some sentences were completely rewritten.

Round 2

Reviewer 2 Report

The authors have made changes according to the recommendations. Thus, I have no further comments.